# The Complexities and Benefits of Community-Partnered Projects for Engineering Capstone Design Students

**Marissa H. Forbes** [1,*]  **and Gordon D. Hoople** [2] 

1    Department of Mechanical Engineering, University of San Diego, San Diego, CA 92110, USA
2    Department of Integrated Engineering, University of San Diego, San Diego, CA 92110, USA
*    Correspondence: mforbes@sandiego.edu

**Abstract:** Community-partnered engineering projects provide a mechanism for cultivating the development of sociotechnical engineers prepared to design within diverse and complex cultural, environmental, social, and other contexts. During the 2021–2022 academic year, we guided three teams of senior undergraduate engineering students through year-long community-partnered projects for their required capstone design course, which instead typically features corporate/industry-sponsored projects. We analyzed end-of-semester reflections (both fall and spring semester) from each student using inductive thematic analysis to explore how they perceived their experiences. The themes that emerged from the student reflections, including connectivity, transdisciplinary, multiple stakeholders, sustainability, justice, and ethics, are all components of the sociotechnical engineering capabilities that we are working to develop in our students. We consider these findings encouraging, and suggestive that integrating community-partnered projects into engineering capstone design offerings is worthwhile and effective. However, our implementation was not without challenges, such as trying to force the projects to fit into a course structure and timeline developed to support corporate/industry-sponsored project teams, which was burdensome to the community-partnered project teams. In this paper, we highlight both the complexities and benefits of this approach and insights gained from student and instructor reflections.

**Keywords:** capstone design; community; engineering; justice; liberal arts; sustainability; transdisciplinary; undergraduate

## 1. Introduction

Engineering is not a technical field. To think of it as such is to erase the why and the who from the problems that engineers are working to solve, and doing so will always lead to harm. Rather, engineering is a sociotechnical field [1] and engineers must consciously design within the cultural, economic, environmental, legal, political, social, etc., context in which the 'problem' is nestled [2]. This is a tall order. To prepare engineers capable of practicing in this multidimensional capacity, to design for a sustainable future, requires a scraping of the traditional, technical engineering education and a thoughtful rebuilding of it. The rebuilding must be people and ecosystem centric. It must be transdisciplinary, integrated, and wholistic.

For decades, many in the engineering higher education space have been working towards this paradigm shift, embracing congruent pedagogical models (such as culturally sustaining pedagogies [3–6] and indigenous-inspired place-based pedagogies [7–10]) and piloting various engineering learning approaches (such as human-centered or user-centered design [11,12]) to facilitate it. Some of these approaches incorporate student–community partnering. For example, service learning (SL) is a widely used pedagogical approach that emphasizes designing with communities and stresses reciprocity in the community partnership [13,14]. The engineering exchange framework [15] builds on this idea and the European 'Science Shop' model that facilitated science–society collaborations by providing

a shopfront where community members could drop in to share needs and/or ideas [16–18]. The engineering exchange framework distances itself from the term 'service', citing its failure to embody the concept of reciprocity, and rather its connotation of a one-directional flow from engineers to community partners, which implicitly reinforces dominant, imperialistic, White power structures [19–24]. Instead, an 'exchange' articulates an equitable partnership between engineers and community partners, with ideas and resources flowing in either direction. This framework also acknowledges multiple ways of knowing and being, especially inspired by Native American and other Indigenous traditions.

Undergraduate engineering degree programs are notoriously burdensome from a courseload perspective; engineering undergraduates are typically required to take more courses than their non-engineering peers on campus [25,26], often extending their degree to four-and-a-half or five years. There is no space in the degree to add new course requirements that will facilitate these sociotechnical, human-centered, sustainability-focused, and community-partnered elements, which must be entwined throughout the modern engineering education. As such, these elements must be the foundation of and integrated into the core engineering courses and electives that the students are already required to take.

In this paper, we present our findings from the development of one such implementation model, through year-long community-partnered engineering design projects for senior undergraduate engineering students as part of their required capstone design course. We highlight both the complexities and benefits of this approach and insights gained from student and instructor reflections from the 2021–2022 academic year.

### 1.1. Institutional Context

The University of San Diego (USD) is a private, Catholic, undergraduate- and liberal arts-focused university serving <10,000 students. The university stresses its commitment to the students' formation of values, to community involvement, and to preparing leaders dedicated to ethical conduct and compassionate service [27].

There are four engineering majors within the engineering school: electrical, industrial and systems, integrated, and mechanical engineering. All engineering students earn a bachelor of science (BS) and bachelor of arts degree, and take the same liberal arts core as their peers studying the arts and sciences. This curricular model is unusual in engineering; typically, American engineering programs require minimal liberal arts courses and culminate in a bachelor of science (BS) degree [25,26]. Each of the engineering programs offered at the university are accredited by ABET, the standard quality assurance organization in the field [28].

### 1.2. Course Context: Engineering Capstone Design

As is standard for ABET-accredited undergraduate engineering programs, students in each major complete an engineering capstone design course. The course serves as a culminating engineering learning experience in which teams of students work on open-ended design projects that require them to integrate and apply their professional and technical skillsets as well as interdisciplinary knowledge. The authors of this paper are instructors for the capstone design course that electrical, integrated, and mechanical engineering students are required to take (the industrial and systems engineering students take a different capstone course).

The course spans a full academic year, including fall and spring semesters that each last ~14 weeks. Each instructor teaches their own section of the course to ~four to six teams, with ~four to six students on each team. Though in separate classrooms and advising different teams, the instructors generally use the same course materials and schedule, and serve as design reviewers for the teams in each other's sections. Each semester, the teams give a formal design review presentation to a panel of faculty and expert consultants who ask challenging questions and provide the team with critical feedback.

*1.3. Course Structure*

The instructors developed the course structure borrowing from the Agile business product development methodology that uses 'sprints', set periods of time during which specified work must be completed and then shared for review [29,30]. The concept is one of rapid iteration, and allows for regular project feedback over short time intervals. In the fall, we define six, two-week sprints that guide students through a first iteration of the engineering design process:

1. Problem Definition,
2. Requirements,
3. Ideation,
4. Analysis,
5. Critical Functioning Prototype, and
6. Detailed Design.

In the spring, we guide the teams to define their own sprints and deliverables, again in two-week increments.

*1.4. Project Types and Selection*

Historically, capstone design projects have been based on design challenges provided by corporate/industry partners, creating a bridge between the academic experience and the professional setting for engineering students about to enter the field. Industry partners usually fund the projects, and can use them as a recruitment mechanism for new hires. In addition to industry projects, USD faculty broadened the project types to also include 1. Entrepreneurship projects (usually self-generated project ideas that require teams to solve both technical and business challenges), 2. competition projects (for student teams wanting to participate in a national or multi-university design competition), and 3. community-focused projects, which are the focus of this paper.

*1.5. Community Projects and Team Members*

During the 2021–2022 academic year, we each taught a capstone design class that was comprised of industry, entrepreneurship, and community projects. There were three community projects across the ten projects in our sections. Below, we provide a brief summary of each community project and an anonymized, alias title that we use to refer to each project throughout this paper.

1. Food production project
   a. Five students (two men, three women): *Andrew, Owen, Laura, Jessica, and Camille.*
   b. Students were tasked with prototyping a small-scale sustainable food production system designed for implementation in a local community experiencing a food desert. The system should produce food and serve as an interactive educational model to teach community members how to replicate the model in their own settings.

2. Youth outreach project
   a. Four students (all men): Rob, Derek, Drew, and Mark.
   b. Students were tasked with creating an engineering project to be used in a youth afterschool setting serving a demographic underrepresented in engineering higher education and practice. The project deliverables included developing kits for students to build remote-controlled robots capable of completing a chosen task, as well as an instruction manual to help facilitators guide students through the engineering learning experience.

3. Waste-upcycling project
   a. Five students (all men): Paul, Jordan, Trevor, Michael, and Eric.
   b. Students were tasked with designing a process and sample products for the production of sellable goods from waste to be used by a youth organization

in Mexico. The team was responsible for designing the workspace layout and creating manuals and a training session for the youth who will be creating the products.

These projects are noticeably different from the more traditional techno-centric industry-sponsored projects which usually task teams with designing small devices, machines, or components. For comparison, some examples of industry projects include designing a device to automatically wash medical instruments, a thermal management system for electronic devices, or developing a portable backup battery system using repurposed batteries.

## 2. Materials and Methods

### 2.1. Student Reflections

Reflection is a critical part of conscientious, responsive, and empathetic engineering practice [31]. We worked to cultivate this skillset in the engineering students through reflection assignments. At the end of each semester, we prompted students to individually complete a reflection, and respond in ~200 words to given prompts. The fall semester reflection prompts were as follows:

- How have you integrated previous coursework from outside of engineering into your capstone design experience?
- Describe your role in detail and how your contributions have impacted the engineering decisions your group made.
- What constraints did you consider and how has it impacted your design?
- How have you approached learning new things when you did not have the knowledge you needed to solve the problem?

The spring semester reflection prompts were as follows:

- What did you learn about working on a team in capstone design that you think will help make you a successful engineer?
- Describe an example of an engineering analysis in your design project that you were personally involved in conducting.
- What societal, ethical, and professional issues did you consider in your capstone design experience?
- Describe an example of an experiment and/or prototype you were personally involved in testing.

### 2.2. Thematic Analysis

We analyzed the fall and spring reflections from each student on the three community-partnered projects using inductive thematic analysis to explore how they perceived their experiences [32]. We use pseudonyms to share quotes from the students in this paper. Although we gave specific reflection prompts, we let themes emerge authentically rather than forcing relation to the original prompt. We identified preliminary themes, then went through an iterative process of mapping the student reflections to the themes, adjusting, and then finalizing the themes.

## 3. Results

We identified three primary themes—'Connectivity', 'Different Than Other Engineering Projects', and 'Justice'—and seven sub-themes. We list the themes and sub-themes in Table 1, and share our findings across the themes in the sections below.

The students also reflected on two additional themes, 'Technical Project Aspects' and 'Teamwork', that we have excluded from this study. Both of these themes are consistent across all engineering capstone design projects, and the purpose of this study is to explore themes unique to the student experiences working on community-partnered projects.

**Table 1.** Identified Themes and Sub-Themes.

| Theme | | Sub-Theme | |
|---|---|---|---|
| 1. | Connectivity | i.<br>ii.<br>iii. | Transdisciplinary<br>Multiple Stakeholders<br>Community |
| 2. | Different Than Other Engineering Projects | i.<br>ii.<br>iii. | Nebulous<br>Qualitative<br>Sustainability |
| 3. | Justice | i. | Ethics |

### *3.1. Connectivity*

In this section, we summarize our findings from the theme Connectivity, and its sub-themes Transdisciplinary (Section 3.1.1), Multiple Stakeholders (Section 3.1.2) and Community (Section 3.1.3). The students saw connections in their projects that extended beyond connections an industry-sponsored team might see of their device or gadget connecting to a larger technical system. These included connections to non-engineering disciplines, to multiple stakeholders with a vested interest in their project outcomes, and to a community (or communities).

### 3.1.1. Transdisciplinary

All engineering capstone projects are designed to be multidisciplinary, as integrating knowledge from various disciplines is a foundational element of the design experience. Typically, students integrate knowledge from their math and science courses, as well as public speaking and writing. However, the students working on the community-partnered projects described having to integrate knowledge from more disciplines, sometimes disciplines they had no experience in.

For example, Andrew worked on the food production project, and relied upon knowledge he gained from "courses such as environmental issues, conceptions of nature, and politics and the environment that discuss the issues of food deserts nationally and food insecurities internationally . . . environmental ethics helped in how [the team] went about [their] project." Andrew's teammate, Owen, stated, "I think that this was a fairly unique project in that many of the techniques and knowledge for the project were totally new." For Owen, "the most important thing that [he] was able to draw from all of [his] classes was the ability to research and put new knowledge into context." Jessica used knowledge from advanced writing, public speaking, and her independent research project. She also, "used information from biology and chemistry classes to figure out what kind of nutrients are needed for both the plants and animals in our system. [She] also used physics to know more about water flow within the system." Capturing this transdisciplinary theme, Jessica wrote:

> *This [project] made me realize how interconnected engineering is with other departments and disciplines.*

On the youth outreach project team, Rob expressed a parallel sentiment and felt that the transdisciplinary nature of his project extended across all disciplines:

> *Almost every single class that I have taken at USD has proven to be beneficial in one way or another for this senior capstone project.*

Rob's teammates Mark and Derek felt similarly. Mark's reflection suggests a migration out of the techno-centric engineering mindset, and into the sociotechnical mindset that we are hoping to cultivate in the students:

> *When thinking about engineering, I always thought about the technical and scientific aspects and situations they undergo and I never really thought about the various other elements that go into engineering.*

Derek saw similarities between the experience of having his beliefs challenged in a theology class and the critical feedback he needed to learn how to receive and address to thrive on his project:

*In my [theology class], I had many of my foundational beliefs regarding politics, religion, and social issues challenged. Not only challenged, but effectively argued against in a way I had not experienced before. Such an experience led me to take uncomfortable criticism and learn how to apply it to strengthen or change my positions, depending on the situation. This lesson was critical for me to take criticism and use it to contribute to the team's effort.*

The transdisciplinary nature of these projects mirrors the trans-dimensional context that sociotechnical engineers must design within, necessitating proficiency spanning cultural, economic, environmental, legal, political, and social realms in addition to their technical fluency. These student reflections indicate that the community-partnered projects provided an authentic experience to cultivate this broad skillset.

### 3.1.2. Multiple Stakeholders

Stakeholders, those parties with a vested interest in a project outcome or who may be impacted by it in some way, are inherent in any engineering design project. For most industry-sponsored projects, there is a single stakeholder: the industry 'sponsor' who the team reports to and who will ultimately be the recipient of the project deliverable(s). Students on the community-partnered projects reflected a different structure, in which each project operated within an ecosystem of multiple stakeholders.

For the waste-upcycling project team, who were collaborating with stakeholders from several organizations in a binational and bilingual setting, communication was critical and challenging. In Jordan's words:

*Our project was a community based project so we ran into many communication issues, not just among ourselves, but with both of the organizations we were working with as well. Not only communication is important, but getting a solid basis of understanding of the topic for the whole group.*

The multiple stakeholders presented complexities and challenges. The team had limited direct communication with the community they were partnered with (two non-profit organization stakeholders directly communicated with the community instead), and Trevor felt like they had to rely on a 'middle-man' rather than getting to interact more directly with the community as they had hoped:

*One of the major constraints we had was the indirect communication between us the engineers and the community we are helping, there was always a middle man and I am not saying that the middle man did not help, I am just saying that direct communication with some members of the community would have been awesome.*

Paul liked having multiple stakeholders and felt like there were more collaborators to go to for help with the project:

*Since our project is a real field project and involved two organizations, that help me to deal with a community issue we faced like shortage of materials and try to find the solutions as soon as possible without spending more money to stay with the budget. Also, it was great experience working with non-profit organization and helping them to reach their goals.*

In addition to the challenge of multiple stakeholders, for the waste-upcycling team, one of those stakeholders was children, which is highly unusual for a traditional capstone project. Michael felt high stakes working towards a design for this user group and, "was concerned with the children who [would] be implementing this process and that it was safe and effective for them and they got strong educational outcomes from it." The youth outreach team had a similar challenge, and Mark described "[having] to take into consideration the possibility that many of these kids will not understand some of the engineering aspects

that go into the production of our [robot]." He went on to describe having to consider the needs of the teachers who would be implementing the engineering activity they designed:

*We also had to take into account that the teachers at [non-profit organization] are not engineers and therefore cannot understand some of the principles and engineering processes of the project as a whole if it is not carefully explained. By knowing and adapting to this, we can create a better and well-rounded product that is usable by anyone.*

This concept of understanding the user, and designing with and for the user to create a better and more accessible solution requires empathy. We posit that these student experiences with multiple stakeholders nurture in them the capability to see from multiple perspectives, which is an essential ingredient for empathy.

### 3.1.3. Community

Perhaps unsurprisingly, the students reflected on community, though they did so in different ways. Paul specified that his project was for a 'real' community, differentiating it from the hypothetical contexts that engineering students are often given. Members of the food production team described connections to the university's neighboring community, where they had initiated relationships. For example, Owen described seeking wisdom from local community members with experience, including the principal of the local middle school:

*Some of the most useful sources were the stories of people who had real experience with [food production systems]: the [local principal] helped us tremendously by giving us access to their now-defunct [food production] project and telling us about the issues that they ran into.*

Camille conveyed hopes for how the implementation of their food production design could benefit the local community:

*A significant portion of our project was aimed towards addressing the societal issue of food insecurity within our local community. While our own system will not be able to feed all the people of [disadvantaged community], our hope is to provide education on the accessibility of [food production systems] to be instituted at things like local churches and schools.*

In other cases, students reflected more generally on how they perceived their projects relating to broader communities. For example, Derek connected his team's youth outreach project to students in underserved communities, and also described a responsibility to 'give back':

*Our project has the potential to be very influential to students in underserved communities who may not have any serious exposure to the [science, technology, engineering, and math] STEM fields. This is also applicable to the ethical context. We had the responsibility to make the best possible product for [the nonprofit organization] to best serve their students and their ability to serve the community as a whole. These issues made us aware of our responsibility to best give back to the students and [the organization].*

Beyond helping to prepare students for professional life, ideally, matriculation through higher education prepares students to be lifelong learners and engaged citizens. It seems that the sense of connection the students felt to a community through their design project provides an opportunity to practice this engagement, which students do not typically experience in their engineering courses.

### 3.2. Different than Other Engineering Projects

In this section, we summarize our findings from the theme Different Thank Other Engineering Projects, and its sub-themes Nebulous (Section 3.2.1), Qualitative (Section 3.2.2) and Sustainability (Section 3.2.3). These community-partnered projects were different than typical engineering projects, and the students felt those differences. Owen summarized the

differences by stating, "Our project had less direct engineering problems than a lot of other projects." Michael felt similarly and noticed his project was less technical, but described more sociotechnical elements:

*I was placed on a community based project compared to a more technical one. I was able to work with and collaborate with many different people all from various different backgrounds. It was nice to be able to meet and figure out how to effectively work with so many different people. I believe that the thing that will make me a successful engineer is my soft skills and my ability to work with virtually anyone-a skill I was able to improve upon in senior design.*

Andrew noticed a difference in terms of timeline, and felt the team needed more time on certain tasks than were allotted for in the course. Though conflicting with the course timeline, he felt their alternative pace was beneficial for the project. He stated:

*Sometimes, I think the slow speed we moved at first is what put us in this place of being open to learn and learning from mistakes along the way to create a better [food production] system in the long run.*

Part of moving 'slow' connected back to the transdisciplinary nature of the project, and was because "none of us had any experience in [food production] and very limited knowledge in cultivating plants and raising aquatic animals. It required a lot of research and being okay with not knowing everything." He reported that the team felt the high stakes of the project, so they were cautious and moved slowly because, "we wanted to go about the right way as we had the health of living beings at the soul of our project."

The youth outreach project team felt their project differed than others in that they had to develop educational components to go along with their technical robot kit design. The team wanted to convey to the youth that engineers do not just tinker or build things, they design for the benefit of humanity. They developed hypothetical design challenges with humanitarian and environmental connections for what the robots could do, such as picking up trash to benefit a given community. In Derek's words:

*Our design challenges had to be adaptable and applicable. A main focus we had was being able to relate engineering to the solving of real time problems. This heavily influenced our decision to proceed with waste collection and humanitarian efforts to be the focus of our design challenges.*

The team had to think about catering the kits and design challenge contexts to specific age groups. Mark reported:

*The main constraint that was considered was the age group of the students that we were working with. The lesson plans and design challenges have also been simplified in order to allow the kids to understand key concepts that went into the ROV, without overwhelming them.*

The waste-upcycling project team felt their project was "a little different than the others in the sense that it was the designing of a process, not a product." (Jordan). This difference made it hard for the group to grasp the project concept in the beginning, and they struggled to articulate it clearly during their first design review. Jordan further described this challenge by stating that, "other groups such as [the competition teams] had a very clear end goal for a product, but for us we had to design an efficient process to get to a product." Paul struggled with the project and feeling like it was well matched to his technical engineering skillset because he felt "it [had] few parts where it need[ed] electrical engineering major students." Despite this struggle, Paul ultimately felt it was worth it:

*[This] project is a real community project that makes it special, also makes it harder in some way but that leads to a huge gain of experience and learning.*

This sense that the differences were 'worth it' because of the gain of experience and learning was an essence conveyed across these student reflections. Though challenging,

there was a sense of appreciation for the opportunity to engage in a project that was perceived as meaningful.

### 3.2.1. Nebulous

Part of the student experience in engaging in these 'different' projects was at times a struggle to grasp the project itself, and the sense that it was a bit nebulous (especially compared to peers working on projects with a clearly defined outcome, such as the design of a technical system component with specific design constraints). Owen described the food production team's challenge in defining their project and the purpose behind it:

*Some of the constraints that we faced were physical and quantifiable, but just as many if not more were based on ideas that we needed our project to embody. We started out with basically no idea what it was that we were going to create or even why, and our constraints reflected this.*

Owen went on to describe the uncertainties as yielding learning opportunities that he could not have gotten in previous experiences where there was a clear answer:

*The lack of direct guidance on what we should actually be creating helped me to learn the importance of developing a real set of objectives and constraints for the project. Obviously I had done this before, but previously there was always one right answer, or they were essentially trivial because I knew exactly what I was going to do anyways.*

Owen's teammate Camille described having to define certainty where they could not originally find it:

*We had to establish our own mode of operation right off the bat along with having to declare our own goals for this project.*

For Camille, the two-week-sprint structure that was imposed on the team for the first semester made this especially challenging because they had to complete deliverables on a timeline that felt irrelevant to their project's best path forward. She expressed:

*This proved to be challenging at first because we had to work using the Sprint model in class, and it felt like we were having to spend way more time making sure we had the right content to turn in to class and get a good grade instead of getting work done that was precisely relevant to our design process and overall project. However, once we entered into [the second semester] and could design our own sprints, things started working out a whole lot better for us. This was when we were able to achieve the tasks that we clearly defined for ourselves. We saved a lot of time only doing what was necessary for discovering what's best for our system, instead of spending time on meeting requirements that weren't exactly applicable to our community project.*

This feeling of having to force the more nebulous community-partnered project to fit into the mold created for the industry-sponsored projects came up numerous times in the student reflections. The general sentiment expressed was that this mold was not a good fit for them or for the project.

### 3.2.2. Qualitative

Related to the nebulous feeling, these students who are well versed in the quantitative realm had to get acquainted with qualitative analyses for these 'different' projects. Andrew described:

*I did all the testing for the small-scale prototype because it was kept at my house. It was interesting because much that came out of this experiment was qualitative results. I was able to learn that qualitative results can be just as informative as quantitative.*

Camille also described engaging in 'atypical' engineering qualitative analyses for the food production project:

*The engineering analysis that I was a part of was the flora and fauna analysis. This wasn't a typical engineering analysis in the sense where a lot of math and science were*

*involved, but instead it involved a lot of research in the details of the key features of each flora and fauna option.*

For the waste-upcycling team, these qualitative efforts included developing flowcharts for the processes they designed and instruction manuals for implementation. After receiving pushback from their design reviewers about their qualitative-only approach, the team also started timing the processes to have quantitative results to share. Eric wrote:

*Since I created the flowcharts for the processes, I was very involved with testing the process to figure out average time to completion. We tested [the process] by running through the process in different ways, and trying new things in order to reduce the time taken to completion. Through this data, the process was reorganized into a more efficient way that moved people around to spaces were (sic) there was help needed.*

The pushback the team received from their reviewers about their qualitative results highlights the expectation that engineering projects should be quantitative. However, sociotechnical engineering values both qualitative and quantitative approaches, and acknowledges that quantitative-only thinking is detrimental to engineering outcomes.

### 3.2.3. Sustainability

The students conveyed an awareness of sustainability beyond that typically seen in industry-sponsored projects. The food production team articulated their efforts to create a microcosmic embodiment of a sustainable cycle in their design. In Andrew's words:

*Our whole context was built around the unjust and unsustainable food system that emphasizes efficiency and economic growth. A more sustainable food system puts the community and environment first to provide communities that cannot afford it a chance to produce their own food. Similarly, it conserves rather than depletes natural resources to ensure environmental health. Having this as the basis of our project really placed emphasis on the social and environmental justice of food production.*

Team members described how this striving for sustainability impacted their design decisions. Owen described the choice to use clear containers for grow beds to provide a learning opportunity for people interacting with the system:

*We also wanted our project to present a new way of looking at agriculture and its many wastes, which is part of why we chose the [name of design]. The clear tubs backfired to some extent, but we still believe that it is important to show a bold, different design to get people interested in sustainable food production.*

Andrew also commented that "everyone agreed on the [design] because it would mimic an ecosystem more which was one of our goals from the beginning." This thought process of prioritizing sustainability in a design is a must for engineers if we are to collectively move towards a sustainable future.

### 3.3. Justice

In this section, we summarize our findings from the theme Justice, and its sub-theme Ethics (Section 3.3.1). The students saw connections between their projects and social and environmental justice. In some cases, the projects served as a vehicle for seeing the engineering field as a whole connecting to justice. For Laura, the project experience made her reconsider pursuing an engineering profession:

*I also learned that engineering can involve social justice issues as well. Before, I did not think I would end up pursuing any sort of career in engineering because I never enjoyed the technical aspects of what was previously required. Once introduced to our project, I was intrigued to not only address modern sustainable food systems through methods such as [food production approach], but also educating what is a very privileged audience to allow them to appreciate more where there food comes from was an important part.*

Rob saw a connection between the youth outreach project and social justice for under-served communities:

*Our entire project revolves around helping underserved communities. Throughout the entire year, we were trying to cater our design to this user group. We did this by trying to make it as affordable as possible and easy to just pick up and learn. We wanted to make a product that was inclusive to all and didn't have any negative impacts on people.*

The waste-upcycling team saw their project connecting to social justice in the context of poverty. Eric shared the same general sentiments that were also expressed by his team members in writing:

*We mainly considered the societal issue of poverty. One of the main goals of our project was to help children in [disadvantaged community] get through high school through the selling of our products. Since we created an effective process that creates products that these students can sell, hopefully this will have a hand in creating economic growth and helping them continue their education.*

We noticed that Eric referred to them as 'our' (as in the team's) products, rather than the community's products, which is a reminder to us that the students were not exposed the concept of community ownership in the project; a critical concept, and yet one that we do not cover in the capstone design course since it is irrelevant to most project teams. We noticed numerous other gaps in the students' understanding of the engineering exchange framework that reflect back to us the shortcomings of the course in preparing the community-partnered project teams to work on and talk about their projects.

### 3.3.1. Ethics

In relation to interfacing with social and environmental injustices, the students were steeped in ethics; both ethical considerations and ethical concerns. Andrew felt strongly about using locally sourced media on the food production grow beds:

*For this project, we wanted to go about the right way and we all were quick to stand up for what we believe in. For example, I would not let lava rocks from Hawai'I be used on our project and I think that is a good way to bring our philosophy and understanding of who we effect (sic) into the project.*

Jessica felt that there were many ethical considerations inherent in working on food production:

*We also addressed the ethical/professional issues of making sure that our system was food safe because we were possibly providing food for people. As we were trying to connect two types of plastic, we needed to make sure that our connection didn't leech into the water, thus adding toxins to the [food that] people would eat. It wouldn't be ethical to provide/sell food that could be toxic to people.*

Trevor felt that he had to apply what he learned in his engineering ethics course to work on the waste-upcycling project:

*I would also say that my engineering ethics course that I have completed has helped me a lot in understanding what an engineer is supposed to be and act in a more professional way, and especially considering our community based project where we are helping a specific community to become a better community.*

The youth outreach team also felt the ethical implications of their work, and it made them cautious in their decision making. Mark described, "we didn't want to hurt anyone's reputation and we wanted to be a role model for the kids at [the organization].'

All engineering has ethical implications, and all responsible designs reflect ethical considerations. However, we were given the sense from the community-partnered project teams that they were called to a higher level of ethical conscientiousness in their work as they saw that work having the potential to impact real people. This differs from most of

the industry-sponsored projects which have more of a hypothetical user that the students are once-removed from, as they only engage with their industry sponsor.

## 4. Discussion

### 4.1. Challenges

The course structure and sprint timelines the instructors developed for the course are a good fit for most industry-sponsored and entrepreneurship projects. This same model, however, is often burdensome to community-partnered projects, like forcing something to progress with a linear trajectory that instead needs to bloom like a flower. For example, our pre-defined sprints and deliverables in the first semester were cumbersome to the community-partnered project teams. In some cases, what we required them to deliver was not a match for what they felt they needed to do to move their projects forward. Though we tried to provide some flexibility for their first-semester sprints, we needed to ensure that their course experience both facilitated their project deliverables while simultaneously meeting the educational objectives by following the engineering design process. The teams were relieved in the second semester, when they got to define their own sprints and deliverables in a way that felt more beneficial to their projects.

Two of the project teams (food production and waste-upcycling) also struggled to perform well in their design reviews—a trend we have seen over multiple years with these types of projects. They had a hard time explaining their non-traditional projects, and the panel reviewers struggled to understand them. The projects came across as more 'soft' or easy, but had to be graded on the same rubric as everyone else. The community-partnered projects may have been less technically complex, but they were also complex in different capacities that were not captured on the rubric or familiar to the instructors.

For example, during their first design review, the waste-upcycling team struggled to describe the 'point' of their project. Their purpose was to design a process, and they were working with several stakeholders across multiple organizations. The review panel pushed for something quantitative, something technical and tangible to come out of the project in order to appropriately grade it. The team ended up designing their process and timing users executing the process to provide a quantitative metric of 'efficiency' against which to gauge their process—and therefore project—success. The feedback from the following design review was favorable; the team was congratulated on their improvements and for having this more tangible and quantitative outcome to show. However, given that ultimately groups of youth will be the ones running the process as they create artistic and innovative products out of waste, is efficiency really the goal? Likely the project outcome and team learning experience would have been better if they were freed from the performative expectation to be like other projects.

But how can we implement a different model for the community-partnered projects when they are just one part of the engineering capstone design requirement? To give community-partnered projects a different, more flexible rubric and project management timeline (i.e., instead of the rigid two-week-sprint structure), perhaps to even group them all in a different course section devoted to the community-design capstone design projects runs a risk of these projects being labelled as a 'soft' and 'easy' version of capstone design. It also communicates that these projects are different than engineering, rather than that they embody the modern sociotechnical engineering paradigm in which all engineering projects should reside. Additionally, the majority of our engineering students do not want to do community-partnered projects for their capstone design experience; most elect industry-sponsored projects or competition projects. Just three out of the ten projects in our sections were community-partnered. However, the teams learn from one another; they are all part of a cohort. Though messier from an implementation standpoint, we think that keeping the different types of project teams together is important, and it allows the other groups to also learn about the community-partnered experience. That said, we do propose to more formally introduce flexibility in defining the first-semester sprints for the community-partnered project teams, and to have the agreed-upon deliverables reflected on

the rubric. We plan to implement these changes and continue to study the experiences of the students participating in community-partnered projects in a longitudinal capacity.

*4.2. Project Preparations*

There is not time during the course to develop a partnership with the community, to engage in the exchange process with the community, and to let the problem identification authentically emerge. This process is essential, but in our experience, it can take years, and cannot be forced into a timeline that fits with the university calendar. As such, we find that these types of projects must be a part of a broader ecosystem of community partnerships in the engineering college or at the university. For example, in partnership with a team of colleagues at USD, we worked to create the Engineering Exchange for Social Justice (ExSJ), a framework for community partnership and co-created, justice-oriented solutions to sociotechnical challenges.

The ExSJ has multiple mechanisms such as summer scholars programs, independent studies, and courses that help with these earlier stages of community partnering, problem identification, and co-ideating solutions with the community. Once the project is fully scoped, then it is ready to be added to the roster of community projects for the year so that a student team can work on designing a solution, ideally with regular input and feedback from the community.

We arrived at this method by trial and error. We learned that putting a project into the capstone design course before it has gone through this process leads to frustrations (for both students and community partners) and poor execution. The waste-upcycling team was one such trial and error. In retrospect, we realize that several more community visits should have taken place, to yield more fine-tuning of the project scope, before having it as a capstone design project. In contrast, the youth outreach project was a multi-year project, born of a multi-year relationship with the community organization. The project was clear, and well defined. As such, the students on this project thrived and had only minor challenges with fitting their project into the course structure and sprint schedule, which we could easily troubleshoot with redefining several of the sprint deliverables.

Student comments about the 'underserved' or 'poor' communities that their projects 'had the potential to be very influential' for or to 'help the community to become a better community' conveyed some of the dominant and savior power structures that we are working to dismantle through our use of the engineering exchange framework. However, students in the capstone design course did not receive any guidance about how to come at the project from the perspective of equitable partnership and exchange rather than 'service.' We recognize that we need to incorporate some of these teachings into the course.

## 5. Conclusions

The themes that emerged from the student reflections, including connectivity, transdisciplinary, multiple stakeholders, sustainability, justice, and ethics, are all components of the sociotechnical engineering capabilities that we are working to cultivate in our students. We consider these findings encouraging, and suggestive that integrating community-partnered projects into engineering capstone design offerings is worthwhile and effective. However, these findings, as well as our own observations from advising the teams, also show that this model is not without challenges.

**Author Contributions:** Conceptualization, M.H.F., G.D.H.; Methodology, M.H.F.; Validation, M.H.F., G.D.H.; Formal Analysis, M.H.F.; Writing—Original Draft Preparation, M.H.F.; Writing—Review & Editing, M.H.F., G.D.H. All authors have read and agreed to the published version of the manuscript.

**Funding:** This research received no external funding.

**Institutional Review Board Statement:** The study was conducted in accordance with the Declaration of Helsinki, and approved by the Institutional Review Board of UNIVERSITY OF SAN DIEGO (IRB-2022-139, 12/1/2021; IRB-2023-336, 3/17/2023).

**Informed Consent Statement:** Informed consent was obtained from all subjects involved in this study.

**Data Availability Statement:** The data are not publicly available due to privacy issues and to ensure confidentiality of the participants.

**Acknowledgments:** The authors would like to thank the students who participated in this study.

**Conflicts of Interest:** The authors declare no conflict of interest.

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
