# Peer review of "The Complexities and Benefits of Community-Partnered Projects for Engineering Capstone Design Students"

_2813-4346, doi:10.3390/higheredu2020016_

Round 1

Reviewer 1 Report

The article presented is very interesting and pertinent. The description of the professional profile of the engineer that is presented is framed in the real needs of this century. Developing multifunctional capabilities that make it possible to design a sustainable future, taking as a working hypothesis the methodology used in the training of these future professionals, from my point of view is a very successful option. And I agree with your purposes, the reconstruction must be centered on people, ecosystems, and this implies that the type of learning must be transdisciplinary, integrated, and holistic.

1º The review of the literature presented in the article is very scarce, taking into account the number of investigations and proposals for systematic reviews carried out internationally on the subject. For future research, a comparative analysis could be carried out with other continents and the determining factors. In any case, the reviewed bibliography is extensive but the lines of work are not reflected in a structured way.

2º The research design is appropriate. It would be interesting to expand the information by designing studies where the research was quantitative or mixed, based on the results presented in the research. The coding system can be improved. It is also necessary to use qualitative analysis software Nvivo or AQUAD or any other that the researchers deem appropriate and that offers us clarity in the analysis carried out.

3º The proposals for improvement or future lines of research are not well defined.

Author Response

Thank you very much for taking the time to review our paper! We appreciate your effort and your feedback. 

  • Comment 1: The review of the literature presented in the article is very scarce, taking into account the number of investigations and proposals for systematic reviews carried out internationally on the subject. For future research, a comparative analysis could be carried out with other continents and the determining factors. In any case, the reviewed bibliography is extensive but the lines of work are not reflected in a structured way.
  • Response: Thank you for this feedback. Though many have explored community projects with engineering students, what we considered unique about this work is the effort to have students engage in the projects within the sociotechnical and engineering exchange frameworks. As such, we sought to focus our literature review on those topics. However, we agree that the suggested comparative analysis would be interesting to pursue for a future study.
  • Comment 2: The research design is appropriate. It would be interesting to expand the information by designing studies where the research was quantitative or mixed, based on the results presented in the research. The coding system can be improved. It is also necessary to use qualitative analysis software Nvivo or AQUAD or any other that the researchers deem appropriate and that offers us clarity in the analysis carried out.
  • Response: We agree with your feedback. Due to the small size of our institution, we typically have ~12-15 capstone teams per year, ~3 of which might be community-partnered projects. Because of these small numbers, and therefore small sample sizes, we have been leery of a quantitative approach for this study. However, perhaps the inclusion of Likert-style surveys and the presentation of our quantitative findings using descriptive statistics only could provide an adequately cautious approach to include quantitative analyses for future iterations. Additionally, perhaps running the study over a number of years would help us gather a larger sample size. We appreciate this suggestion and will apply it to our future research planning. Similarly, because of the small sample size, we carried out the coding ‘by hand’, but agree with your assessment that our coding system can be improved, and can implement your suggestion in the future.
  • Comment 3: The proposals for improvement or future lines of research are not well defined.
  • Response: Thank you for this feedback. We added some text to the discussion section to outline some changes we plan to make to the course and our plans for continued study in a longitudinal capacity.

Reviewer 2 Report

Thank you for the opportunity to read this manuscript. While the concept of community-partnered engineering projects has been explored in previous research, this particular study appears to be novel in its focus on the development of sociotechnical engineering capabilities in undergraduate engineering students through community partnerships, as well as in its analysis of student reflections to explore their perceptions and experiences. The study also highlights the challenges and benefits of integrating community-partnered projects into a course structure traditionally designed for corporate/industry-sponsored projects.

 Regarding the Methods section, I wonder if the authors could address some questions that emerged as I was reading the draft and perhaps incorporate them into their section:

 How did the use of sprints and rapid iteration impact the students' ability to complete their community-partnered projects?

Were there any challenges in implementing the Agile methodology in an academic setting, and if so, how were they addressed?

How did the use of sprints and regular feedback impact the quality of the students' work and their overall learning experience?

Did the students have any input in the development of the course structure, and if so, how did that impact their engagement and motivation?

How does this course structure compare to other capstone design courses in terms of student learning outcomes and project success rates?

 As for the Discussion:

 I would like to see the authors expand their section by addressing the following points:

 In the authors' view, how could the course structure be adapted or modified to better accommodate community-partnered projects while still maintaining the same level of rigor and assessment criteria?

 How can the rubric and assessment criteria be revised to better capture the unique complexities and challenges of community-partnered projects, based on the project outcomes?

 Finally, while the results section presents some interesting findings, it can be quite elaborate. To make it more reader-friendly, it may be helpful to include 2-3 diagrams or visual aids to better visualize the findings both conceptually and in terms of content analysis.

Author Response

Thank you very much for reviewing our paper! We appreciate your efforts and your feedback.

  • Comment 1: Regarding the Methods section, I wonder if the authors could address some questions that emerged as I was reading the draft and perhaps incorporate them into their section:
    • How did the use of sprints and rapid iteration impact the students' ability to complete their community-partnered projects?
    • Were there any challenges in implementing the Agile methodology in an academic setting, and if so, how were they addressed?
    • How did the use of sprints and regular feedback impact the quality of the students' work and their overall learning experience?
    • Did the students have any input in the development of the course structure, and if so, how did that impact their engagement and motivation?
      • Response: Because these questions (and our responses to them are related), we will respond to them as a group. We found that the use of sprints facilitated a rapid deliverable/feedback process that was beneficial to all teams. Although we don’t have a control for true comparison, our sense is that both the quality of their work and their overall learning experience benefited from applying this methodology.

In the first semester, we defined the sprints and deliverables for the students, but they get to define their sprints and deliverables during the second semester (we outlined this in section 1.3 of the paper).

Our pre-defined sprints and deliverables in the first semester were cumbersome to the community-partnered project teams. In some cases, what we required them to deliver was not a match for what they felt they needed to do next to move their project forward.

In terms of students having input in the development of the course structure, they do in the sense that they get to chart their own course in the second semester. In general, we found students to take this responsibility seriously, and saw it impact their engagement and motivation.

In general, we found the implementation of the Agile methodology to be very beneficial and successful in the academic setting. The students liked it and took to it organically. The only challenges were with the community-partnered projects, as outlined above.

We have added some text to the Discussion section to elaborate on these points.

  • How does this course structure compare to other capstone design courses in terms of student learning outcomes and project success rates?
    • Response: This is a great question and something we are curious about too. However, because of the small sample size from our course we can’t answer this question yet. Perhaps in the future we will be able to get a large enough sampling to answer this question in a longitudinal study.

  • Comment 2: As for the Discussion:  I would like to see the authors expand their section by addressing the following points:
    • In the authors' view, how could the course structure be adapted or modified to better accommodate community-partnered projects while still maintaining the same level of rigor and assessment criteria?
    • How can the rubric and assessment criteria be revised to better capture the unique complexities and challenges of community-partnered projects, based on the project outcomes?
      • Response: We will respond to these questions together since they are related. These are important points. As we mentioned in paragraph 3 of section 4.1, we feel it is critical to keep the course structure largely as-is, despite some of the challenges. That said, we also think it’s important to formalize the process of providing flexibility in defining first-semester sprints and deliverables for the project teams, and to have those changes reflected on the rubrics. We have added text to the paper to help make this point.
    • Finally, while the results section presents some interesting findings, it can be quite elaborate. To make it more reader-friendly, it may be helpful to include 2-3 diagrams or visual aids to better visualize the findings both conceptually and in terms of content analysis.
      • Response: Thank you for this feedback. Though we like the idea of including diagrams/visual aids, we don’t think there is a visual that would convey helpful information in the results section other than what we provided in Table 1. However, we added a sentence at the beginning of each theme to help organize and to help the reader orient themselves to which themes/subthemes are in that section.